



# The Impact of Aerosol on Cloud Water:
# A Heuristic Perspective

Fabian Hoffmann[1], Franziska Glassmeier[2], and Graham Feingold[3]

[1]Ludwig-Maximilans-Universität München, Meteorologisches Institut, Munich, Germany
[2]Delft University of Technology, Delft, Netherlands
[3]Chemical Sciences Laboratory, NOAA, Boulder, Colorado

**Correspondence:** Fabian Hoffmann (fa.hoffmann@lmu.de)

**Abstract.** Aerosol-cloud interactions modulate the role of clouds in Earth's climate. We derive, evaluate, and apply a simple model to understand aerosol-mediated cloud water adjustments in stratocumulus based on only two prognostic equations for the integrated cloud water $L$ and droplet number concentration $N$. The model is solved numerically and analytically, and agrees well with documented large-eddy simulation data and satellite retrievals. A tight relationship between adjustments at low and

high $N$ is found, revealing the influence of non-precipitation processes (primarily entrainment) on adjustments in precipitating clouds. Furthermore, it is shown that adjustments in non-precipitating clouds tend to be positively biased by external $L$ or $N$ perturbations, while adjustments in precipitating clouds are barely susceptible. By deliberately reducing the complexity of the underlying system, this study constitutes a way forward to facilitate process-level understanding of cloud water adjustments.

## 1 Introduction

By constituting the nuclei on which cloud droplets form, aerosol substantially shapes the microphysical composition of clouds, their optical properties, and hence their role in Earth's climate. One important example is the ability of clouds to reflect incident solar radiation back to space, causing a negative (cooling) influence on Earth's radiation budget. While aerosol tends to increase cloud reflectance, this and other aerosol-cloud-climate interactions are only marginally understood (e.g., Boucher et al., 2013; Forster et al., 2021).

One metric to quantify aerosol-cloud-climate interactions is the susceptibility of the shortwave cloud albedo $A$ to changes in the droplet concentration $N$ (e.g., Platnick and Twomey, 1994). This susceptibility can be expressed as

$$S \equiv \frac{\mathrm{d}\ln(A)}{\mathrm{d}\ln(N)} = \frac{1-A}{3}\left[1 + \frac{5}{2}\frac{\mathrm{d}\ln(L)}{\mathrm{d}\ln(N)}\right], \tag{1}$$

where the term $(1-A)/3 \geq 0$ represents the fairly well understood increase of $A$ with $N$ at constant cloud water, commonly referred to as the *Twomey effect* (Twomey, 1974, 1977). This study will address the considerably less understood cloud water

adjustments $\mathrm{d}\ln(L)/\mathrm{d}\ln(N)$ in the bracketed term. Depending on how the vertically integrated cloud water $L$ changes with $N$, cloud water adjustments can increase, decrease, or even change the sign of $S$.

    In earlier years, cloud water adjustments were thought mainly to be related to precipitation suppression, i.e., the increasingly less efficient production of precipitation by smaller cloud droplets, resulting in larger $L$ for higher $N$, causing a larger $S$ than





anticipated from the Twomey effect alone (e.g., Albrecht, 1989). Later, it was recognized that the mixing of clouds with their
environment (entrainment) increases for higher $N$, which causes $L$ to decrease, resulting in a smaller or even negative $S$ (e.g.,
Wang et al., 2003; Ackerman et al., 2004; Bretherton et al., 2007; Glassmeier et al., 2021). Together, these effects result in an
*inverted v* dependency of $L$ on $N$, and thus a commensurately more nuanced influence of cloud water adjustments on $S$, which
has been retrieved from satellite observations of shallow cumulus and stratocumulus clouds (e.g., Gryspeerdt et al., 2019).

Large-eddy simulation (LES) has become the primary tool to gain process-level understanding of cloud water adjustments.
While LES estimates stem from the high-resolution representation of the underlying dynamics and cloud microphysics, they
tend to be valid only for limited spatial domains and specific initial and boundary conditions (e.g., Ackerman et al., 2009;
Glassmeier et al., 2021). On the other hand, satellite observations have become increasingly useful for an integrated view of
aerosol-cloud-climate interactions, sampling a wealth of real-world data, but also their inherent co-variabilities that confound
process-level understanding (e.g., Gryspeerdt et al., 2019; Mülmenstädt et al., 2024).

Aiming to combine the aforementioned integrated view with process-level insights, this letter will develop a heuristic model
for cloud water adjustments in stratocumulus, a crucial cloud type in Earth's radiation budget (e.g., Wood, 2012). The foun-
dations of this model will be laid out in Sec. 2, and it will be applied in Sec. 3. Basic sensitivities to model parameters are
analyzed in Sec. 4. Section 5 addresses the variability of cloud water adjustments in externally perturbed systems, presenting
a way forward to use this study's results when interpreting observed cloud water adjustment. The letter is summarized and
40 concluded in Sec. 6.

## 2 Model Formulation

The heuristic model is formulated using ideas that originated from satellite retrievals by Gryspeerdt et al. (2019) and the LES
modeling by Hoffmann et al. (2020): Their works showed that cloud water adjustments can be separated into two distinct
regimes, which are dominated by precipitation at low $N$, and *thermodynamics* at high $N$, respectively. Here, *thermodynamics*
comprises the effects of entrainment, radiative cooling, and surface fluxes on $L$. Thus, we will refer to the underlying processes
in those regimes as driven by *precipitation* or *thermodynamics* in the following. Moreover, we introduce the shorthand

$$m \equiv \frac{\mathrm{d}\ln(L)}{\mathrm{d}\ln(N)} \tag{2}$$

for the change in $L$ with $N$. The optional subscripts h and l indicate limits for high and low $N$, respectively. The $\infty$ subscript
marks (potentially prescribed) steady states.

The effects of precipitation and thermodynamics on the temporal change in $L$ are represented as

$$
\begin{aligned}
\frac{\mathrm{d}L}{\mathrm{d}t} &= -c_1 \frac{L^{3/2}}{N} + \frac{L_{\infty,\mathrm{h}}(N) - L}{\tau_{\mathrm{t}}} \\
&= -\frac{2}{3} \frac{L}{\tau_{\mathrm{p}}(L,N)} + \frac{L_{\infty,\mathrm{h}}(N) - L}{\tau_{\mathrm{t}}},
\end{aligned}
\tag{3}
$$

whose terms will be described next.





The first term on the right-hand-side of (3) represents a precipitation sink. The employed expression relates the cloud base

rain rate to $L$ and $N$. We express the precipitation sink by introducing a precipitation timescale

$$\tau_{\mathrm{p}} = \left| \frac{\mathrm{d}}{\mathrm{d}L} \left( -c_1 \frac{L^{3/2}}{N} \right) \right|^{-1} = \frac{2}{3} \frac{1}{c_1} \frac{N}{L^{1/2}}, \tag{4}$$

where the term in parentheses is a more common representation of the cloud base rain rate, which has been assessed observa-
tionally and theoretically (e.g., Van Zanten et al., 2005; Kostinski, 2008). It has been argued that the exponents of $L$ and $N$
depend on the assumed sedimentation velocity and hence droplet size (Feingold et al., 2013). For simplicity, these dependencies

are neglected here, as is evaporation below cloud base.

The second term on the right-hand-side of (3) depicts the charge/discharge to the thermodynamic carrying capacity of the
system $L_{\infty,\mathrm{h}}$, which can also be interpreted as a steady state $L$ at high $N$ whose existence has been discussed by Hoffmann
et al. (2020). For a given $N$, this term can be a sink to the $L$ budget due to an excess in entrainment warming and drying
($L > L_{\infty,\mathrm{h}}$), or a source driven by longwave radiative cooling ($L < L_{\infty,\mathrm{h}}$) [cf. Fig. 2 in Hoffmann et al. (2020)]. The timescale

associated with this process is given by $\tau_{\mathrm{t}}$. The thermodynamic carrying capacity is expressed as

$$L_{\infty,\mathrm{h}} = L_0 \left( \frac{N}{N_0} \right)^{m_{\infty,\mathrm{h}}}, \tag{5}$$

where $m_{\infty,\mathrm{h}}$ determines how $L_{\infty,\mathrm{h}}$ changes with $N$, while $L_0$ and $N_0$ are constant parameters.

Many studies have determined $m_{\infty,\mathrm{h}}$, and hence $L_{\infty,\mathrm{h}}$, for high $N$ only to exclude the effects of precipitation present at low
$N$. Here, $L_{\infty,\mathrm{h}}$ is applied to all $N$ with the same $m_{\infty,\mathrm{h}}$. This idea is motivated by the insight that the temporal change in $L$ due

to thermodynamics (entrainment, radiative cooling, and surface fluxes) exhibits a sensitivity to $N$ that seems independent of
the presence of precipitation. This was initially shown in Fig. 3 of Hoffmann et al. (2020), but is recreated in a more useful way
in the supplement (Fig. S1). Thus, the the same adjustment of thermodynamic processes at high $N$ (i.e., $m_{\infty,\mathrm{h}}$) are assumed
to persist for low $N$. Specifically, $m_{\infty,\mathrm{h}} < 0$ due to the increase in entrainment with $N$ (Wang et al., 2003; Ackerman et al.,
2004; Bretherton et al., 2007).

The primary model parameters $\tau_{\mathrm{t}} = 9\,\mathrm{h}$, $c_1 = 7600\,\mathrm{m}^{-2}\,\mathrm{kg}^{-1/2}\,\mathrm{s}^{-1}$, $L_0 = 90\,\mathrm{g\,m}^{-2}$, $N_0 = 100\,\mathrm{cm}^{-3}$, and $m_{\infty,\mathrm{h}} = -0.64$
have been chosen to match the ensemble LES modeling of Glassmeier et al. (2021), who studied cloud water adjustments
in stratocumulus clouds, and should be seen as one potential realization of cloud water adjustments. The sensitivity to these
parameters will be analyzed in Sec. 4. Note that to fit the aforementioned ensemble LESs, $c_1$ is about half the value observed by
Van Zanten et al. (2005), necessary to account for the subadiabaticity of $L$ naturally included in observations, but not captured

in (3). Further, note that the thermodynamic charge/discharge in (3) is driven by the *linear* difference $L_{\infty,\mathrm{h}} - L$, without
further justification. A model driven by the *logarithmic* difference $\ln(L_{\infty,\mathrm{h}}) - \ln(L)$ does not align well with the ensemble
LES modeling of Glassmeier et al. (2021), but is briefly discussed in the supplement (Text S1 and Fig. S2).

For completeness, a prognostic equation for $N$, loosely based on Baker and Charlson (1990), is solved. The expression

$$\frac{\mathrm{d}N}{\mathrm{d}t} = c_2 N \left( -c_1 \frac{L^{3/2}}{N} \right) - c_3 N^2 + S_{\mathrm{N}} \tag{6}$$

combines sinks of $N$ by precipitation (first term on the right-hand-side) and Brownian coagulation (second term), as well as a
source $S_{\mathrm{N}}$ (third term) that could represent, e.g., the emission of sea spray. Here, we choose $c_3 = 10^{-15}\,\mathrm{m}^3\,\mathrm{s}^{-1}$ (e.g., Seinfeld





and Pandis, 2016) and $c_2 = 3\,\mathrm{m^2\,kg^{-1}}$, which can be considered the upper limit for $c_2$ (Wood, 2006). Nonetheless, the effect of precipitation scavenging steered by $c_2$ on the steady state behavior of $L$ is small, as we will show next.

## 3 An Initial Assessment

Results from integrating (3) and (6) for 7 days with a timestep $\Delta t = 1\,\mathrm{min}$ are shown in Fig. 1a. The source $S_\mathrm{N}$ has been neglected for simplicity. In total, 250 simulations are conducted, with initial $L$ and $N$ randomly placed between $1\,\mathrm{g\,m^{-2}}$ and $1000\,\mathrm{g\,m^{-2}}$, as well as $1\,\mathrm{cm^{-3}}$ and $100\,000\,\mathrm{cm^{-3}}$, while only results for $N \leq 10\,000\,\mathrm{cm^{-3}}$ are shown. Note that while $N < 10\,\mathrm{cm^{-3}}$ are frequently observed in stratocumulus, they tend to not exhibit $N > 1000\,\mathrm{cm^{-3}}$ (e.g., Wood, 2012). This discrepancy is irrelevant to this study that focuses on the change of $L$ with $N$, i.e., the slope $m$, which is constant for such high 95   $N$, as we will see below.

    The individual simulations (gray lines in Fig. 1a) show substantial motion in the $L$ direction, while motion in the $N$ direction is only visible for low $N$ due to precipitation scavenging and high $N$ due to Brownian coagulation. Ultimately, the trajectories of these simulations approach a steady state $L_\infty$ (brown dots) that agrees well with the ensemble LES reference by Glassmeier et al. (2021) (black line), and especially its slopes $m_\mathrm{h}$ and $m_\mathrm{l}$ toward high and low $N$. As we will show below, these slopes 100   agree well with the heuristic model's steady state slopes $m_{\infty,\mathrm{h}} = -0.64$ and $m_{\infty,\mathrm{l}} = 0.24$ (red and blue lines, respectively). The only notable difference to the LES reference is the more gentle transition between the two slopes, which might be due to the continuous representation of precipitation in (3), while the process of autoconversion, i.e., the initiation of precipitation, is a discontinuous process that only allows precipitation to form once a certain droplet size is exceeded (Kessler, 1969). This threshold is illustrated by the dashed black line showing a cloud top effective droplet radius of $14\,\mu\mathrm{m}$ that is often used to 105   discriminate precipitating from non-precipitating clouds (e.g., Gerber, 1996).

    Without $N$ dynamics, the steady state $L_\infty$ exhibits very similar features (Fig. 1b). Most importantly, the slopes and hence the cloud water adjustments agree, which is why $N$ dynamics are neglected in the following. A reason for the apparent independence of cloud water adjustments from $N$ dynamics is shown in Fig. 1c, which shows the relative motion of the system, $|[\mathrm{d}\ln(L)/\mathrm{d}t]/[\mathrm{d}\ln(N)/\mathrm{d}t]| = |\mathrm{d}\ln(L)/\mathrm{d}\ln(N)]|$. Relative changes in $L$ exceed changes in $N$ almost everywhere in the phase 110   space (warm colors). Changes in $N$ dominate primarily around the steady state (cold colors), where $\mathrm{d}L/\mathrm{d}t = 0$ per definition. Brownian coagulation widens this region around the steady state for high $N$, while precipitation scavenging creates another region where $N$ dynamics dominate at low $N$ but for $L \gg L_\infty$. Although precipitation scavenging is often reported for low $N$ with potential implications for cloud water adjustments (e.g., Gryspeerdt et al., 2022), the $L_\infty$ investigated here are too small to be affected by stronger $N$ dynamics, thus allowing us to neglect them for now. Future work might want to include a prognostic 115   equation for the cloud fraction, which tends to be smaller than unity for low $N$, resulting in higher in-cloud $L$ than predicted by (3) and thus stronger precipitation scavenging by (6).





To further understand the steady state behavior of $L$ and its slope $m$, we investigate $\mathrm{d}L/\mathrm{d}t = 0$ of (3) analytically. A few algebraic rearrangements yield

$$L_\infty = L_{\infty,\mathrm{h}}\left(1 + c_1\tau_\mathrm{t}\frac{L_\infty^{1/2}}{N}\right)^{-1} = L_{\infty,\mathrm{h}}\left(1 + \frac{2}{3}\frac{\tau_\mathrm{t}}{\tau_\mathrm{p}}\right)^{-1}. \tag{7}$$

The term in parentheses describes the deviation of $L_\infty$ from $L_{\infty,\mathrm{h}}$ due to precipitation, and its strength depends on the ratio of the process timescales $\tau_\mathrm{t}$ and $\tau_\mathrm{p}$. Figure 1d shows the $N$ dependence of $\tau_\mathrm{t}$, $\tau_\mathrm{p}$, and the timescale of all $L$ processes, $\tau_\mathrm{L} = \left(\tau_\mathrm{t}^{-1} + \tau_\mathrm{p}^{-1}\right)^{-1}$ in the steady state. While $\tau_\mathrm{t}$ (long-dashed red line) is constant as prescribed, a strong increase in $\tau_\mathrm{p}$ (short-dashed red line) with $N$ is shown, indicating that precipitation affects $L_\infty$ only for sufficiently small $N$. Thus, $\tau_\mathrm{L}$ (continuous red line) follows $\tau_\mathrm{p}$ for low $N$ and $\tau_\mathrm{t}$ for high $N$, while $N \approx 100\,\mathrm{cm}^{-3}$ can be considered the boundary between
the precipitation- and thermodynamics-dominated parts of the phase space. $\tau_\mathrm{L}$ from the ensemble of LESs of Glassmeier et al. (2021) (black line) captures this behavior only partially, which might be related to the difficulty in determining multiple derivatives from LES data.

The logarithmic derivative of (7) with respect to $\ln(N)$ gives

$$m_\infty = \frac{m_{\infty,\mathrm{h}}}{1 + \frac{\tau_\mathrm{t}}{\tau_\mathrm{p}}} + \frac{\frac{2}{3}\left(m_{\infty,\mathrm{h}} + 1\right)}{1 + \frac{\tau_\mathrm{p}}{\tau_\mathrm{t}}}, \tag{8}$$

with more details provided in the supplement (Text S2). $m_\infty$ shows that for $\tau_\mathrm{t} \ll \tau_\mathrm{p}$ ($N \gg 100\,\mathrm{cm}^{-3}$), thermodynamics dominate cloud water adjustments via $m_{\infty,\mathrm{h}}$. For $\tau_\mathrm{p} \ll \tau_\mathrm{t}$ ($N \ll 100\,\mathrm{cm}^{-3}$), $m_\infty$ approaches

$$m_{\infty,\mathrm{l}} = \frac{2}{3}(m_{\infty,\mathrm{h}} + 1), \tag{9}$$

which combines the effects of thermodynamic adjustments, $m_{\infty,\mathrm{h}}$, with a precipitation adjustment of $2/3$. This behavior is captured well in Figs. 1a and b, where the slopes $m_{\infty,\mathrm{h}} = -0.64$ (red line) and $m_{\infty,\mathrm{l}} = 0.24$ (blue line) overlap with the model
data.

Additionally, the relationship (9) constitutes a way to assess the consistency of cloud water adjustments derived for low and high $N$. Strictly speaking, the $m_{\infty,\mathrm{h}}$ derived from $m_{\infty,\mathrm{l}}$ via (9) only represents the thermodynamic adjustments at low $N$, while the thermodynamic adjustments at high $N$ might differ. Nonetheless, the cloud water adjustments of $m_\mathrm{l} = 0.21$ and $m_\mathrm{h} = -0.64$ determined from the ensemble of LESs by Glassmeier et al. (2021) agree well with (9). Deviations from (9) can
indicate aerosol-meteorology co-variability commonly found in maritime and continental air masses (e.g., Brenguier et al., 2003), but absent in the LESs of Glassmeier et al. (2021) by design. Moreover, deviations can hint to changes in the sensitivity of thermodynamic processes to $N$, e.g., via precipitation-induced decoupling of the cloud layer, which is more prevalent at low $N$, which can result in significant reduction in the supply of moisture from the surface (e.g., Nicholls, 1984; Hoffmann et al., 2023).





## 4 Sensitivity to Model Parameters

Now, the dependence of model (3) on the parameters $\tau_\text{t}$, $c_1$, $L_0$, and $m_{\infty,\text{h}}$ is tested in Figs. 2a to d, showing $L$ after 7 days of integration as a function of $N$. The dependence on $N_0$ is neglected here, as it is analogous to $L_0$ via its influence on $L_{\infty,\text{h}}$. If $\tau_\text{t}$, $c_1$, $L_0$, or $m_{\infty,\text{h}}$ are not varied, their aforementioned defaults are used. The default case is indicated by gray dots, while setups with varied parameters are highlighted by colored dots. Timestepping and initialization follow the previously outlined procedure.

Figure 2a shows that shorter $\tau_\text{t}$ force $L$ to follow $L_{\infty,\text{h}}$ for lower $N$ compared to the default case. The commensurately higher $L$ at low $N$ is caused by a faster recharge of precipitation losses by thermodynamics, while $L$ at high $N$ is unchanged. As expected from (8), $m_\text{h}$ and $m_\text{l}$ approach the slopes $m_{\infty,\text{h}}$ and $m_{\infty,\text{l}}$ for all $\tau_\text{t}$. However, the transition between $m_\text{h}$ and $m_\text{l}$ is shifted depending on the ratio $\tau_\text{p}/\tau_\text{t}$. A similar influence is visible from variations in the precipitation constant $c_1$, which determines the strength of precipitation losses at small $N$ (Fig. 2b). Note that the value of $c_1$ closest to the observations by Van Zanten et al. (2005) (yellow dots) results in stronger precipitation losses than in the ensemble LESs of Glassmeier et al. (2021) (gray dots).

$L$ changes proportionally to $L_0$ for all $N$, with its slopes matching $m_{\infty,\text{l}}$ and $m_{\infty,\text{h}}$ as before (Fig. 2c). For all $L_0$, the maximum $L$ agrees well with the cloud top effective droplet radius of $14\,\mu\text{m}$ (dashed line), marking the transition between precipitating and non-precipitating clouds. Note that this is not the case for the previously discussed sensitivities on $\tau_\text{t}$ and $c_1$ (Figs. 2a and b). This indicates that the usefulness of the cloud top effective radius threshold for separating the precipitating and non-precipitating branches of $L$ depends on $\tau_\text{t}$ and $c_1$.

The sensitivity to $m_{\infty,\text{h}}$ is displayed in Fig. 2d. As indicated by (8), the slopes for high and low $N$ are commensurate with the prescribed values $m_{\infty,\text{h}}$ and the resultant $m_{\infty,\text{l}}$. Nonetheless, we would like to highlight a few interesting values that $m_{\infty,\text{h}}$ may assume. For $m_{\infty,\text{h}} = 2.0$ (dark blue dots), cloud water adjustments are the same for all $N$ ($m_{\infty,\text{l}} = m_{\infty,\text{h}}$), marking the limit for the previously discussed *inverted v* cloud water adjustments. Consequently, any $m_{\infty,\text{h}} > 2.0$ will result in *regular v* cloud water adjustments. Coincidentally, $m_{\infty,\text{h}} = 2.0$ matches the slope of the effective radius (dashed line). If $m_{\infty,\text{h}} = -1.0$ (orange dots), cloud water adjustments vanish at low $N$ ($m_{\infty,\text{l}} = 0$), while they vanish at high $N$ for $m_{\infty,\text{h}} = 0.0$ (light blue dots). Sufficiently strong negative cloud water adjustments can offset the Twomey effect and thus cause a decrease in cloud albedo with increasing $N$, i.e., $S < 0$ according to (1). Obviously, $m_{\infty,\text{h}} < -0.4$ (green dots) causes negative $S$ for high $N$, but $m_{\infty,\text{h}} < -1.6$ (brown dots) establishes negative $S$ for all $N$ by also guaranteeing that $m_{\infty,\text{l}} < -0.4$.

## 5 A Perturbed System

Building on the previous analysis of the unperturbed steady state behavior of the model (3), we now like to understand its suesceptibility to external perturbations in $N$ and $L$. $N$ perturbations exist at various temporal and spatial scales, covering highly localized aerosol emissions such as ship tracks to phenomena on regional scales like volcanic eruptions. At the same time, these perturbations might exhibit correlations with $L$.



In this study, perturbations are modeled as a Bernoulli process, making it possible to impose perturbation timescales $\tau_{\mathrm{prt}}$ between $20\,\mathrm{min}$ and $2\,\mathrm{weeks}$. If a perturbation takes place, $\ln(N)$ is modified by adding a $\Delta\ln(N)_{\mathrm{prt}} = \xi\sigma_{\mathrm{prt}}$, where $\xi$ is a normally distributed random number with zero mean and unity standard deviation, modified by $\sigma_{\mathrm{prt}} = 0.5, 1.0$, or $2.0$. At the same time, a $\Delta\ln(L) = m_{\mathrm{prt}}\Delta\ln(N)_{\mathrm{prt}}$ is added to $\ln(L)$, with $m_{\mathrm{prt}} = -1.0, 0.0$, or $1.0$ to introduce correlations in the perturbation. Note that $\tau_{\mathrm{prt}}$, $\sigma_{\mathrm{prt}}$, and $m_{\mathrm{prt}}$ are chosen to elucidate the general sensitivity of the system, and not to match a realistic case. However, the resultant variability is similar to satellite retrievals (e.g., Gryspeerdt et al., 2019) if sufficiently slow perturbations are applied (cf. Fig. S3). We use the default model parameters described above. Timestepping and initialization follow the previously outlined procedure. No $N$ dynamics other than the perturbation are considered. Results are averaged over the last $2\,\mathrm{days}$ of the $7\,\mathrm{days}$ simulations. In total, $100\,000$ simulations are executed for each configuration.

Figures 3a to c show example distributions of $L$ and $N$ for $m_{\mathrm{prt}} = -1.0, 0.0$, and $1.0$, respectively, with the same $\sigma_{\mathrm{prt}} = 1.0$ and $\tau_{\mathrm{prt}} = 0.3\,\mathrm{h}$ for all cases. The short $\tau_{\mathrm{prt}}$ has been chosen to highlight some processes that are more subtle at larger $\tau_{\mathrm{prt}}$ (see also Fig. S3 for $\tau_{\mathrm{prt}} = 10\,\mathrm{h}$). For high $N$, one sees that the variability in $L$ for a given $N$ is proportional to the difference between $m_{\mathrm{prt}}$ and $m_{\infty,\mathrm{h}}$, which determines the time required for thermodynamics to counter a perturbation in $L$. Because thermodynamic charge/discharge is linear in $L$ and the perturbations are applied in $\ln(L)$ space, more time is required to deplete a positive $L$ perturbation than a negative. This asymmetric response results in slightly higher mean $L$ (black lines) than in the unperturbed case, and adjustments *appear* less negative than the prescribed ($m_{\mathrm{h}} > m_{\infty,\mathrm{h}}$).

A similar effect is also visible for low $N$. As long as $L$ is sufficiently affected by the perturbation ($m_{\mathrm{prt}} = -1.0$ and $1.0$, Figs. 3a and c), larger $L$ are possible due to the aforementioned asymmetric response by thermodynamics to the perturbation. However, precipitation removes positively perturbed $L$ more efficiently for lower $N$ than for larger, causing adjustments to appear more positive than in the unperturbed case ($m_{\mathrm{l}} > m_{\infty,\mathrm{l}}$). For perturbations in $N$ only ($m_{\mathrm{prt}} = 0.0$, Fig. 3b), the mean $L$ increases more gently than in the unperturbed cases, which results in $m_{\mathrm{l}} < m_{\infty,\mathrm{l}}$. This is due to stronger precipitation for negative $N$ perturbations, removing any excess in $L$ more quickly than thermodynamics can increase $L$ for positive $N$ perturbations.

Figure 3d shows $m_{\mathrm{l}}$ (blue lines) and $m_{\mathrm{h}}$ (red lines) as a function of $\tau_{\mathrm{p}}$. The slopes have been determined by linear regression from the mean $\ln(L)$, using the respective ranges $1\,\mathrm{cm}^{-3} < N < 5\,\mathrm{cm}^{-3}$ and $1\,000\,\mathrm{cm}^{-3} < N < 10\,000\,\mathrm{cm}^{-3}$, which have been chosen to minimize the influence of the transition zone between the slopes.

The strongest impact of perturbations on $m_{\mathrm{h}}$ (red lines) is visible for $m_{\mathrm{prt}} = 1.0$ and it scales with $\sigma_{\mathrm{prt}}$ as one would expect. Interestingly, all tested perturbations cause $m_{\mathrm{h}} > m_{\infty,\mathrm{h}}$, but it is expected that more strongly negative perturbations ($m_{\mathrm{prt}} \ll -1.0$) could cause a $m_{\mathrm{h}} < m_{\infty,\mathrm{h}}$. Nonetheless, the influence of perturbations vanishes for $\tau_{\mathrm{prt}} \gg \tau_{\mathrm{t}}$, i.e., when thermodynamic charge/discharge becomes faster than the perturbation. Similarly, $m_{\mathrm{l}}$ (blue lines) is not affected when $\tau_{\mathrm{prt}} \gg \tau_{\mathrm{p}}$. Because $\tau_{\mathrm{p}} \ll \tau_{\mathrm{t}}$ for low $N$ (cf. Fig. 1d), $m_{\mathrm{l}}$ is much less susceptible to perturbations than $m_{\mathrm{h}}$. Overall, $m_{\mathrm{l}}$ is closer to $m_{\infty,\mathrm{l}} = 0.24$ than $m_{\mathrm{h}}$ to $m_{\infty,\mathrm{h}} = -0.64$ for most tested configurations. Thus, $m_{\mathrm{l}}$ might constitute a way to constrain the unperturbed $m_{\infty,\mathrm{h}}$ via (9), while $m_{\mathrm{h}}$ might not necessarily enable conclusions on $m_{\infty,\mathrm{h}}$ as long as perturbations cannot be ruled out.





## 6 Summary and Conclusions

Understanding aerosol-cloud interactions is crucial for constraining the effects of aerosols on the climate. In this study, a heuristic model to understand aerosol-mediated cloud water adjustments in stratocumulus has been derived, evaluated, and applied. The model has been developed to predict the evolution of cloud water path $L$ as a function of the cloud droplet number concentration $N$. Although the concurrent evolution in $N$ can have an impact on the evolution of $L$ (e.g., Gryspeerdt et al., 2022), it has been neglected for most of this study, and $N$ has been considered a mere parameter. The reason for this is that the relatively small steady state $L$ to which the system converged does not enable substantial changes in $N$ by precipitation scavenging.

For the evolution of $L$, two processes have been considered: (i) the removal of $L$ by precipitation, and (ii) changes in $L$ by *thermodynamics*, i.e., the integrated effect of entrainment, radiation, and surface fluxes. The analytical and numerical analysis of the prognostic equation for $L$ shows that it represents the development of two distinct slopes $m = \mathrm{d}\ln(L)/\ln(N)$. One is dominated by precipitation at low $N$ and the other by thermodynamics at high $N$, which is in agreement with previous studies using satellite retrievals (e.g., Gryspeerdt et al., 2019) and large-eddy simulations (LESs) (e.g., Hoffmann et al., 2020; Glassmeier et al., 2021). The study finds that these slopes are intimately related via

$$m_\mathrm{l} = \frac{2}{3}(m_\mathrm{h} + 1),$$

showing that precipitation adjustments at low $N$, $m_\mathrm{l}$, are partially controlled by the thermodynamic adjustments dominating at high $N$, i.e., $m_\mathrm{h}$. Thus, this relationship implicitly assumes the same thermodynamic adjustments $m_\mathrm{h}$ for all $N$.

The slopes determined from an ensemble of LESs (Glassmeier et al., 2021) obey the aforementioned relationship between $m_\mathrm{l} = 0.21$ and $m_\mathrm{h} = -0.64$ well. However, this LES ensemble did not include aerosol-meteorology co-variability by design, and hence justifies the use of the same thermodynamic adjustments $m_\mathrm{h}$ for all $N$. Observed values for $m_\mathrm{l}$ are between $0.1$ and $0.4$ (e.g., Christensen and Stephens, 2011; Gryspeerdt et al., 2019; Possner et al., 2020), which would require $m_\mathrm{h}$ to vary between $-0.9$ and $-0.4$ to follow the aforementioned relationship. But these values only barely overlap with the observed range for $m_\mathrm{h}$ between $-0.2$ and $-0.4$ (e.g., Christensen and Stephens, 2011; Gryspeerdt et al., 2019; Possner et al., 2020). This discrepancy indicates stronger thermodynamic adjustments at low $N$ that transition into weaker thermodynamic adjustments at high $N$, suggesting that $m_\mathrm{h}$ should be a function of $N$.

Aerosol-meteorology co-variability could be an explanation for this $N$ dependency. However, we would like to emphasize that this aerosol-meteorology co-variability does not have to be exogenous [e.g., differences in continental and maritime air (e.g., Brenguier et al., 2003)], but could be created by the analyzed system endogenously [e.g., the decoupling of the cloud deck at low $N$, restricting the supply of moisture from the surface (e.g., Nicholls, 1984; Hoffmann et al., 2023)]. Quantifying the influence of aerosol-meteorology co-variability on the relationship between $m_\mathrm{l}$ and $m_\mathrm{h}$ constitutes an interesting way to continue this study, and to deepen process-level understanding of aerosol-cloud-climate interactions.

Another explanation for the weaker observed $m_\mathrm{h}$ are external perturbations affecting $N$ and $L$. Our results show that thermodynamic adjustments are sensitive to perturbations with timescales of a few tens of hours or less, causing $m_\mathrm{h}$ to be weaker than in unperturbed simulations, i.e., to be closer to the aforementioned observations, while $m_\mathrm{l}$ is barely affected (cf. Fig. 3d).



To constrain the role of aerosols and clouds in the climate system, these perturbations and their biases have to be given
due consideration. At the same time, eliminating the effects of perturbations is similarly important for a deeper process-level
understanding of cloud water adjustments. Simple models like the one developed here seem to be a useful approach to condense
the wealth of theoretical, modeling, and observational knowledge gained so far. Together, this strengthens the need to combine
top-down and bottom-up approaches to advance our understanding of aerosol-cloud-climate interactions (e.g., Mülmenstädt
and Feingold, 2018; Glassmeier et al., 2019).

*Data availability.*  The data to reproduce Figs. 1 to 3 is archived in a repository (Hoffmann et al., 2024).

*Author contributions.*  FH, FG, and GF conceived the study. FH carried out the study and has written the initial manuscript. FG and GF
revised the manuscript.

*Competing interests.*  GF and FG are co-editors of ACP. Other than that, the authors declare that they have no competing interests.

*Acknowledgements.*  FH appreciates support from the Emmy Noether program of the German Research Foundation (DFG) under grant
HO 6588/1-1. FG acknowledges support from The Branco Weiss Fellowship - Society in Science, administered by ETH Zurich, and by
the European Union (ERC, MesoClou, 101117462). Views and opinions expressed are however those of the author(s) only and do not
necessarily reflect those of the European Union or the European Research Council Executive Agency. Neither the European Union nor the
granting authority can be held responsible for them. GF acknowledges funding from NOAA's ERB program (NOAA CPO Climate and CI
03-01-07-001).





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






**Figure 1.** Panel (a) and (b) show trajectories of individual simulations (gray lines) in an $L$-$N$ phase space with and without $N$ dynamics, respectively. Brown dots indicate the location of simulations after $7$ days. The quotient of the relative motion in $L$ and $N$ directions is shown in panel (c). These panels are overlayed with the corresponding ensemble LES reference by Glassmeier et al. (2021) (thick black line), the slopes $m_{\infty,l} = 0.24$ and $m_{\infty,h} = -0.64$ (blue and red lines, respectively), and the $14\,\mu m$ cloud top effective droplet radius (black dashed line). Panel (d) shows the process timescales $\tau_t$, $\tau_p$, and $\tau_L$ (red lines), as well as ensemble LES reference by Glassmeier et al. (2021) (black line).







**Figure 2.** $L$ after $7$ days as a function of $N$ for variations in (a) $\tau_t$, (b) $c_1$, (c) $L_0$, and (d) $m_{\infty,h}$ (colored dots). The default configuration is differentiated by gray dots. Plots are overlayed with $m_{\infty,l} = 0.24$ and $m_{\infty,h} = -0.64$ (blue and red lines), and the $14\,\mu\text{m}$ cloud top effective droplet radius (black dashed line).





**Figure 3.** Joint $L$-$N$ histograms (opaque colors) and mean $\ln(L)$ (thick black line) for perturbations in $L$ and $N$ for $\tau_{\mathrm{prt}} = 0.3\,\mathrm{h}$, $\sigma_{\mathrm{prt}} = 1.0$ with (a) $m_{\mathrm{prt}} = -1$, (b) 0.0, and (c) 1.0. Plots are overlayed with $m_{\infty,\mathrm{l}} = 0.24$ and $m_{\infty,\mathrm{h}} = -0.64$ (blue and red lines), and the $14\,\mu\mathrm{m}$ cloud top effective droplet radius (black dashed line). Note that the histograms are normalized such that the integral over each $N$ column yields 1 (cf. Gryspeerdt et al., 2019). Panel (d) shows the fitted slopes $m_{\mathrm{l}}$ (blue lines) and $m_{\mathrm{h}}$ (red lines) for $\sigma_{\mathrm{prt}} = 0.5$ (thin lines), 1.0 (medium lines), 2.0 (thick lines), and $m_{\mathrm{prt}} = -1.0$ (dashed lines), 0.0 (continuous lines), 1.0 (dashdotted lines).