# Peer review of "The Impact of Aerosol on Cloud Water: A Heuristic Perspective"

_EGUsphere, 2024_

## Referee Comment (RC1)

Review of " The Impact of Aerosol on Cloud Water: A Heuristic Perspective" by Hoffmann et al. [Research Article, egusphere-2024-1725]

This paper employed a heuristic model, which was derived from two prognostic equations (liquid water path $L$ and cloud droplet number concentration $N$), to understand cloud water adjustments in stratocumulus to aerosols. The primary model parameters were chosen by matching the ensemble LES modeling of Glassmeier et al. (2021). This heuristic model successfully reproduced the inverted "v" shape relationship for $L$-$N$, which was found in LES simulations and satellite retrievals: $L$ increases with aerosols at low $N$ via suppressing precipitation, while $L$ decreases with aerosols at high $N$ through thermodynamic effects such as entrainment drying. Intriguingly, the authors found a tight relationship between adjustments at low and high $N$, demonstrating that entrainment effects that predominate at high $N$ influence adjustments in precipitating clouds at low $N$. They also examined the sensitivity of cloud water adjustments in precipitating and non-precipitating clouds to the heuristic model's parameters, along with external $L$ or $N$ perturbations.

I enjoyed reading this paper. It was very well organized and easy to follow. This study showcased a useful and effective way of applying a simple heuristic model to decipher the intricate cloud-aerosol interactions (ACI). The tight relationship between adjustments at high and low $N$ found within this study is also beneficial in identifying potential aerosol-meteorology co-variability in the ACI study. I believe this paper will be suitable for publication in ACP if some issues outlined below are addressed.

***Major comments:***

1. Lines 69-70 (or L69-70): The authors claimed that $L_{\infty,h}$ is applied to all $N$ with *the same $m_{\infty,h}$*, because they assumed that the sensitivity of temporal change in $L$ due to thermodynamics to $N$ *seems* not dependent on the presence of precipitation. However, if we take a closer look at Figure S1 (or see figure below), the sensitivity of $L$ tendency due to entrainment (see the slope of the green line) to $N$ is found to be notably different between the precipitation period and the non-precipitation period.

[Figure]

*Figure R1. Same as Figure S1, but adding black dashed lines to help discern the difference in green line slopes between the presence (marked by green areas) and non-presence (marked by pink areas) of precipitation.*

Specifically, when precipitation occurs (or at low $N$), the boundary layer becomes more stable, thereby leading to a relatively small sensitivity of $L$ tendency due to entrainment to $N$. Conversely, the sensitivity should be relatively large when precipitation is absent (or at high $N$), as demonstrated by Figure R1. Given these facts, the authors need to clarify the rationality of assuming $m_{\infty,h}$ is independent of $N$ and discuss the impact of this hypothesis on the main conclusion.

The above sensitivity contrast also indicates weaker thermodynamic adjustments (mainly entrainment) at low $N$ that transition into stronger thermodynamic adjustments at high $N$, which is, however, opposite to the authors' discussion in L230-234. Are there any reasons for the inconsistency?

2. In Section 5, the authors examined the susceptibility of cloud water adjustments to external perturbations in $N$ and $L$. They modeled these perturbations as Bernoulli processes. The perturbation of $\tau$, $\sigma$, and $m$ is chosen to represent the general sensitivity of the system rather than matching a realistic case. I am curious if it is possible to perturb these parameters or impose $N$ ($L$) perturbations per the influence of large-scale meteorological factors (MFs) like the moisture contrast between 1000 hPa and 700 hPa, which can alter the efficiency of entrainment drying and thus influence cloud water adjustments, especially at high $N$. Such a perturbation due to MFs would be more realistic and physically reasonable.

3. The paper is well-structured and concise, but in certain places, it is overly brief, particularly when introducing concepts without sufficient explanation. This brevity may stem from the text limitations imposed by the previous submission to GRL. Given that ACP does not have such restrictions, I recommend that the authors expand on and clarify specific concepts or physical mechanisms in more detail. Below, I provide some examples for the authors' consideration.

(a) L45: The authors claimed that thermodynamic effects on $L$ include the influence of entrainment, radiative cooling, and surface fluxes. It would be helpful if the authors could elaborate on how these three terms affect cloud water at a process level.

(b) Suggest briefly explaining the concepts of "entrainment warming and drying", "Brownian coagulation", "Bernoulli process", etc., and adding citations as well.

4. L31: The authors mentioned the limitations of LESs in understanding cloud water adjustments due to limited spatial domains and specific initial and boundary conditions. However, the authors tuned their heuristic model parameters to match the ensemble LES modeling of Glassmeier et al. (2021). In that regard, I'd assume the heuristic model derived here is subject to LESs' limitations. Generally, this study would be more insightful if the authors could use one paragraph or section to discuss their model's limitations and possible improvements (e.g., including a prognostic equation for cloud fraction), helping refine its applicability in future research.

***Minor comments:***

L16: "droplet concentration" to "cloud droplet number concentration"

L33: Did you mean co-variability of aerosols and meteorology? Please be specific.

L35: "letter" to "paper"

L39: "letter" to "paper"

L64: Does this source refer to cooling-induced water vapor condensation or enhanced PBL turbulent moistening?

L68: Add references for "many studies"

L72: Remove a duplicate "the"

L75: It would be helpful if the authors could provide more technical details on tuning the parameters to align with ensemble LES modeling. Additionally, including a validation figure of $L$ evolution predicted by the heuristic model relative to LES modeling would enhance the clarity and robustness of the study.

L104: Please clarify how the cloud top effective droplet radius was derived and plotted in Figure 1.

L124: Is the threshold of 100 for $N$ consistent with findings from LES modeling or satellite observations?

L163-171: The authors highlighted some interesting values when perturbing $m_{\infty,h}$. However, I am not sure if $m_{\infty,h} = 2.0$ is physically meaningful as entrainment is supposed to dominate at high $N$ (Figure S1), and the sinking term (entrainment) for $L$ outweighs the source terms (longwave radiative cooling and surface fluxes), yielding $m_{\infty,h} < 0$. It might be better to take into account some physical constraints when perturbing the parameters.

---

## Author Comment (AC1)

Response to Reviewer 1

This paper employed a heuristic model, which was derived from two prognostic equations (liquid water path $L$ and cloud droplet number concentration $N$), to understand cloud water adjustments in stratocumulus to aerosols. The primary model parameters were chosen by matching the ensemble LES modeling of Glassmeier et al. (2021). This heuristic model successfully reproduced the inverted "v" shape relationship for $L$-$N$, which was found in LES simulations and satellite retrievals: $L$ increases with aerosols at low $N$ via suppressing precipitation, while $L$ decreases with aerosols at high $N$ through thermodynamic effects such as entrainment drying. Intriguingly, the authors found a tight relationship between adjustments at low and high $N$, demonstrating that entrainment effects that predominate at high $N$ influence adjustments in precipitating clouds at low $N$. They also examined the sensitivity of cloud water adjustments in precipitating and non- precipitating clouds to the heuristic model's parameters, along with external $L$ or $N$ perturbations.

I enjoyed reading this paper. It was very well organized and easy to follow. This study showcased a useful and effective way of applying a simple heuristic model to decipher the intricate cloud-aerosol interactions (ACI). The tight relationship between adjustments at high and low $N$ found within this study is also beneficial in identifying potential aerosol-meteorology co-variability in the ACI study. I believe this paper will be suitable for publication in ACP if some issues outlined below are addressed.

*We thank the reviewer for the support of our work and the constructive comments, which helped to clarify various aspects of our manuscript.*

**Major comments:**

1. Lines 69-70 (or L69-70): The authors claimed that $L_{\infty,h}$ is applied to all $N$ with *the same $m_{\infty,h}$*, because they assumed that the sensitivity of temporal change in $L$ due to thermodynamics to $N$ *seems* not dependent on the presence of precipitation. However, if we take a closer look at Figure S1 (or see figure below), the sensitivity of $L$ tendency due to entrainment (see the slope of the green line) to $N$ is found to be notably different between the precipitation period and the non- precipitation period.

*We agree with this observation. However, while one can easily determine d(dL/dt|$_{entrainment}$)/dN from Fig. S1, one would have to assume a timescale to estimate m = dln(L)/dln(N). And this timescale might not be constant, as indicated by $\tau_L$ in Fig. 1d. Thus, it is hard to translate any variability seen (or not seen) in Fig. S1 into cloud water adjustments. Hence, we decided to follow the established notion that cloud water adjustments can be represented by two constant slopes for low and high N, respectively.*

Specifically, when precipitation occurs (or at low N), the boundary layer becomes more stable, thereby leading to a relatively small sensitivity of L tendency due to entrainment to N. Conversely, the sensitivity should be relatively large when precipitation is absent (or at high N),

as demonstrated by Figure R1. Given these facts, the authors need to clarify the rationality of assuming $m_{\infty,h}$ is independent of N and discuss the impact of this hypothesis on the main conclusion.

*We would like to repeat a part of our conclusions: "This discrepancy indicates stronger thermodynamic adjustments at low N that transition into weaker thermodynamic adjustments at high N , suggesting that $m_h$ should be a function of N." This exactly corresponds to the reviewer's comment above. Interestingly, the derivation of (9), i.e., our main theory, can be done for any piecewise constant $m_{\infty,h}$. Thus, our theory allows one to use different $m_{\infty,h}$ for low and high N. We have emphasized this in the conclusions: "Note that any piecewise-constant $m_h$ obeys the aforementioned relationship with $m_l$, making it possible to use different $m_h$ for low and high N in the proposed framework."*

The above sensitivity contrast also indicates weaker thermodynamic adjustments (mainly entrainment) at low N that transition into stronger thermodynamic adjustments at high N, which is, however, opposite to the authors' discussion in L230-234. Are there any reasons for the inconsistency?

*As outlined above, a weaker $d(dL/dt|_{entrainment})/dN$ at low N does not necessarily indicate weaker $m_{\infty,h}$ at low N, as we need to consider the role of an unknown timescale. In fact, a stronger $m_{\infty,h}$ at low N is a reasonable assumption as evaporating precipitation stabilizes the boundary layer and hence decreases entrainment. Thus, precipitation suppression due to an increase in N will allow turbulence to spin up and increase entrainment. We have clarified this in our conclusions: "Aerosol-meteorology co-variability could be an explanation for this N dependency. However, we would like to emphasize that this aerosol-meteorology co-variability does not have to be exogenous [e.g., differences in continental and maritime air (e.g., Brenguier et al., 2003)], but could be created by the analyzed system endogenously [e.g., the stabilizing effect of evaporating precipitation on boundary-layer dynamics and hence entrainment (e.g., Nicholls, 1984; Wood, 2007; Hoffmann et al., 2023)]."*

2. In Section 5, the authors examined the susceptibility of cloud water adjustments to external perturbations in N and L. They modeled these perturbations as Bernoulli processes. The perturbation of $\tau$, $\sigma$, and $m$ is chosen to represent the general sensitivity of the system rather than matching a realistic case. I am curious if it is possible to perturb these parameters or impose N (L) perturbations per the influence of large-scale meteorological factors (MFs) like the moisture contrast between 1000 hPa and 700 hPa, which can alter the efficiency of entrainment drying and thus influence cloud water adjustments, especially at high N. Such a perturbation due to MFs would be more realistic and physically reasonable.

*This is an important comment as it addresses the usefulness of the heuristic model to answer specific questions. Since the heuristic model does not represent entrainment explicitly, it cannot be used to answer a question on entrainment directly. Since we know that a decrease (increase) in entrainment efficiency results in an increase (decrease) in L, we can use the heuristic model to assess the influence of L perturbations, as done in Sec. 5. To address these specific questions,*

*especially on entrainment, more directly, we are currently developing a slightly more complex approach, in which entrainment-related processes are depicted more directly. We hope to submit the results in the coming months.*

3. The paper is well-structured and concise, but in certain places, it is overly brief, particularly when introducing concepts without sufficient explanation. This brevity may stem from the text limitations imposed by the previous submission to GRL. Given that ACP does not have such restrictions, I recommend that the authors expand on and clarify specific concepts or physical mechanisms in more detail. Below, I provide some examples for the authors' consideration.

    (a) L45: The authors claimed that thermodynamic effects on $L$ include the influence of entrainment, radiative cooling, and surface fluxes. It would be helpful if the authors could elaborate on how these three terms affect cloud water at a process level.

        *We slightly lengthened our descriptions of this: "The first term on the right-hand-side of [dL/dt] represents a precipitation sink." and "For a given N, this term can be a sink to the L budget due to an excess in entrainment warming and drying causing the cloud to evaporate ($L > L_{\infty,h}$), or a source driven by longwave radiative cooling leading to more condensation ($L < L_{\infty,h}$), while the effect of surface fluxes is usually small [cf. Fig. 2 in Hoffmann et al. (2020)]."*

    (b) Suggest briefly explaining the concepts of "entrainment warming and drying", "Brownian coagulation", "Bernoulli process", etc., and adding citations as well.

        *In the revised version of the manuscript, we state that "entrainment warming and drying caus[es] the cloud to evaporate." We, however, would like to refrain from explaining established concepts in too much detail because additional explanations could affect the readability of the text. Nonetheless, to respond to the other reviewer's comment, we slightly extended our explanation of the 'Bernoulli process' by "In this study, perturbations are modeled as a Bernoulli process, and are applied with the probability $\Delta t/\tau_{prt}$ evaluated for every timestep of the model. Here, $\tau_{prt}$ is the perturbation timescale, which is varied from 20 min to 2 weeks."*

4. L31: The authors mentioned the limitations of LESs in understanding cloud water adjustments due to limited spatial domains and specific initial and boundary conditions. However, the authors tuned their heuristic model parameters to match the ensemble LES modeling of Glassmeier et al. (2021). In that regard, I'd assume the heuristic model derived here is subject to LESs' limitations. Generally, this study would be more insightful if the authors could use one paragraph or section to discuss their model's limitations and possible improvements (e.g., including a prognostic equation for cloud fraction), helping refine its applicability in future research.

*This is again an important point, and we agree with it. In fact, we already discussed that the model and hence the LES data do not agree with observational data, indicating that different*

*m$_{\infty,h}$ need to be considered for low and high N: "This discrepancy [of the heuristic model and observations] indicates stronger thermodynamic adjustments at low N that transition into weaker thermodynamic adjustments at high N , suggesting that m$_h$ should be a function of N." While our conclusions section briefly addresses the LESs lack of aerosol-meteorology co-variability ("[...] this LES ensemble did not include aerosol-meteorology co-variability by design [...]") and external perturbations ("Another explanation for the weaker observed m$_h$ are external perturbations affecting N and L."), this study is not the right place for a full assessment of potential shortcomings in LESs, especially the impact of numerics which could impact the representation of entrainment. Thus, we already stated in the original version of the manuscript "that this set of [LES] parameters should be seen as one potential realization of cloud water adjustments", indicating that the parameters used here need to be adapted for any future application. Lastly, the idea of including a prognostic equation for cloud fraction is appealing, and could play an important role for the consideration of aerosol-meteorology co-variability. Thus, our manuscript states that "[q]uantifying the influence of aerosol-meteorology co-variability on the relationship between m$_l$ and m$_h$ constitutes an interesting way to continue this study [...]", while not detailing how one could do so.*

**Minor comments:**

L16: "droplet concentration" to "cloud droplet number concentration"

*Done.*

L33: Did you mean co-variability of aerosols and meteorology? Please be specific.

*Yes. Changed to "[...] but also the inherent co-variability of aerosol and meteorology that confounds process-level understanding [...]"*

L35: "letter" to "paper"

*Done.*

L39: "letter" to "paper"

*Done.*

L64: Does this source refer to cooling-induced water vapor condensation or enhanced PBL turbulent moistening?

*The primary way is more condensation of water vapor due to cooler temperatures. While we agree that increased turbulence increases the surface moisture flux, its impact seems negligible for the clouds analyzed here (see Fig. S1). We clarified this as "For a given N, this term can be [...] a source driven by longwave radiative cooling leading to more condensation (L < L$_{\infty,h}$), while the effect of surface fluxes is usually small [cf. Fig. 2 in Hoffmann et al. (2020)]."*

L68: Add references for "many studies"

*We added a reference to Fig. 1 in Glassmeier et al. (2021), in which a selection of these studies is presented.*

L72: Remove a duplicate "the"

*Done.*

L75: It would be helpful if the authors could provide more technical details on tuning the parameters to align with ensemble LES modeling. Additionally, including a validation figure of *L* evolution predicted by the heuristic model relative to LES modeling would enhance the clarity and robustness of the study.

*We have added some further information: "The model parameters have been chosen to match the ensemble LES modeling of Glassmeier et al. (2021), who studied cloud water adjustments in stratocumulus clouds. They determined $\tau_t$ = 9 h and $m_{\infty,h}$ = −0.64 using an emulator. Based on their Fig. 3a, we selected $L_0$ = 90 g $m^{-2}$ and $N_0$ = 100 $cm^{-3}$ to match their $L_\infty$ for high N , and derived $c_1$ = 7600 $m^{-2}$ $kg^{-1/2}$ $s^{-1}$ to match their $L_\infty$ for low N." Assessing the evolution of L is interesting but not within the scope of this study, which focuses on steady-state solutions.*

L104: Please clarify how the cloud top effective droplet radius was derived and plotted in Figure 1.

*Simple adiabatic considerations show that the cloud top effective radius scales with $(L/N^2)^{1/6}$, as shown in Goren et al. (2019), which is now referenced in the manuscript. We adapted the following statement: "This threshold is illustrated by the dashed black line indicating a cloud top effective droplet radius of 14 μm that is often used to discriminate precipitating from non-precipitating clouds, and scales with $(L/N^2)^{1/6}$ (e.g., Gerber, 1996; Goren et al., 2019)."*

L124: Is the threshold of 100 for *N* consistent with findings from LES modeling or satellite observations?

*This value is not universal, but it matches the $L_\infty$ inflection point of this and other studies well. We added the following statement: "Note that we introduce N ≈ 100 $cm^{-3}$ as the boundary between the precipitation- and thermodynamics-dominated and hence low and high N parts of the phase space, as it corresponds to the $L_\infty$ inflection point in the heuristic model and LES ensemble data of Glassmeier et al. (2021) (Figs. 1a and b)."*

L163-171: The authors highlighted some interesting values when perturbing $m_{\infty,h}$. However, I am not sure if $m_{\infty,h}$ = 2.0 is physically meaningful as entrainment is supposed to dominate at high *N* (Figure S1), and the sinking term (entrainment) for *L* outweighs the source terms

(longwave radiative cooling and surface fluxes), yielding $m_{\infty,h} < 0$. It might be better to take into account some physical constraints when perturbing the parameters.

*We agree that $m_{\infty,h} > 0$ is unphysical. However, we want to explore the possibilities of the phase space. Thus, we caution the reader as follows: "Nonetheless, we would like to highlight a few interesting values that $m_{\infty,h}$ may assume, even though $m_{\infty,h} > 0$ is likely unphysical due to the negative impact of increased entrainment on L at higher N."*

---

## Author Comment (AC2)

Response to Reviewer 2

This study heuristically devised a simple model describing the behavior of liquid water path (L) as a function of cloud droplet number concentration (N) with its parameters adjusted to previously performed LES simulations and perturbed to explore the system sensitivity. The simple model is also used to analyze how the system behaves depending on external forcing of L and N. As the main framework of this study, these analyses are performed in terms of the logarithmic sensitivity parameter of L with respect to N where the precipitation and thermodynamics controls are identified and linked with each other. This is a very interesting study that provides a useful process-level insight into the cloud water adjustment to aerosol perturbations. I only have some relatively minor comments (listed below) that mainly require further clarifications of some model setup. I would recommend the manuscript be published after these comments are appropriately addressed.

*We thank the reviewer for the support of our study, and the comments that helped to remedy unclear parts of the manuscript.*

Specific points:

Line 27, Line 166: What is "inverted v"? I cannot find the description of "v".

*The 'inverted v' refers to the shape of L caused by increase for lower N, followed by a decrease for higher N. However, we have removed any references to 'inverted v' in the revised manuscript.*

Line 166: Likewise above, what is "regular v"?

*With 'regular v', we are refereeing to the case in which L increases for lower N, followed by a stronger increase for higher N. However, we have removed the sentence mentioning 'regular v' in the revised manuscript.*

Line 65: Insert "Based on (2)" prior to "The thermodynamic carrying capacity is...".

*Following the reviewer's suggestion, we changed the sentence to: "The thermodynamic carrying capacity is derived from (2), and expressed as [...]"*

Line 90: "The source $S_N$ has been neglected for simplicity": Does this mean that N is monotonically decreasing with time during the time integration according to (6)? If so, N should not reach the steady state. Please explain what happens with temporal evolution of N in this computational setup.

*Yes, N is monotonically decreasing. We amended the revised manuscript as follows to address the reviewer's comment: "The individual simulations (gray lines in Fig. 1a) show substantial motion in the L direction, while motion in the N direction is only relevant at low N < 100 cm$^{-3}$*

*due to precipitation scavenging and at high $N > 1000$ cm$^{-3}$ due to Brownian coagulation. Although $S_N = 0$ and hence $dN/dt < 0$ everywhere in the phase space, a stable population of simulations persists between these limits for at least the 7 days of simulated time considered (brown dots). [Baker and Charlson (1990) showed how the consideration of a $S_N > 0$ could offset the losses in $N$, causing $N_\infty$ steady states.] In the L direction, these simulations approach a steady state $L_\infty$ [...].”*

Line 106: “Without N dynamics”: Does this mean that only (3) is used without (6)? Please clarify.

*Exactly. We have clarified the revised manuscript: “Solving only (3), i.e., without the N dynamics considered by (6), the steady state $L_\infty$ exhibits very similar features to the previously discussed solution (Fig. 1b).”*

Line 177: Would it be possible to write down the equation describing how the perturbation timescale ($t_{prt}$) comes into the perturbation added ($\Delta \ln(N)_{prt}$). It is unclear (at least for me) how the prescribed perturbation timescale is used in calculation of the perturbation.

*This was indeed unclear and has been revised as: “In this study, perturbations are modeled as a Bernoulli process, and are applied with the probability $\Delta t/\tau_{prt}$ evaluated for every timestep of the model. Here, $\tau_{prt}$ is the perturbation timescale, which is varied from 20 min to 2 weeks.”*

---

## Author Response (AR2)

Response to Reviewer 1

Thanks for the authors' responses. The authors have addressed most of my concerns very well. However, one issue remains from the last point of my first major comment: "The above sensitivity contrast also indicates weaker thermodynamic adjustments … Are there any reasons for the inconsistency?"

We thank the reviewer for the additional comments. We will address the reviewer's previous major comment with our answer to the following comment.

I am still unclear about the assumption of a stronger $m_{\infty,h}$ at low N. In the authors' responses, they mentioned that at low N, evaporating precipitation stabilizes the boundary layer, thereby weakening entrainment. Based on this, I would expect that reduced entrainment would lead to a weaker thermodynamic adjustment or $m_{\infty,h}$. Could the authors provide further clarification on their assumption at low N?

Two things are to consider. First, we will address the negative slope of $m_{\infty,h}$ at low N. Entrainment is proportional to the integrated buoyancy flux (Nicholls and Turton 1987). Evaporating precipitation negatively contributes to the buoyancy flux. Thus, entrainment decreases as precipitation increases. This stabilizing effect of precipitation has been outlined in Caldwell et al. (2005) and Wood (2007) before. Since precipitation and hence the potential for its evaporation decreases with increasing N, entrainment increases with N, which decreases L with N, and results in a negative $m_{\infty,h}$. Note that this is very similar to the previously described sedimentation- or entrainment feedbacks which increase entrainment with increasing N (Bretherton et al. 2007, Wang et al. 2003). Second, we must consider that precipitation is only present for sufficiently low N < 100 cm$^{-3}$. For higher N, there is no precipitation and no evaporation to stabilize the boundary layer. Thus, the $m_{\infty,h}$ from evaporating precipitation is only restricted to low N, while the $m_{\infty,h}$ from sedimentation- and evaporation-entrainment feedback might exist for all N. Combining these two facts, the $m_{\infty,h}$ at low N should be stronger (more negative) at low N than at higher N. We extended our manuscript as: "Moreover, deviations can hint at changes in the sensitivity of thermodynamic processes to N, e.g., the stabilizing effect of evaporating precipitation on boundary-layer dynamics and hence entrainment at low N, which naturally vanishes for higher N due to decreasing precipitation (e.g., Nicholls and Turton 1984; Caldwell et al. 2005; Wood 2007; Hoffmann et al. 2023)."

Additionally, I suggest that the authors include a table of heuristic model parameters, listing their names and physical meanings, to enhance the readability of the paper.

This is a great suggestion. We added the table.